# Graphdiyne and Nitrogen-Doped Graphdiyne Nanotubes as Highly Efficient Electrocatalysts for Oxygen Reduction Reaction

**DOI:** 10.3390/ijms242316813

**Published:** 2023-11-27

**Authors:** Tongchang Liu, Xinmeng Hao, Jiaqi Liu, Pengfei Zhang, Jiaming Chang, Hong Shang, Xuanhe Liu

**Affiliations:** School of Science, China University of Geosciences (Beijing), Beijing 100083, China; 1006210228@cugb.edu.cn (T.L.); hxm206121@163.com (X.H.); 1006210227@cugb.edu.cn (J.L.); 1006210230@cugb.edu.cn (P.Z.); 1006210225@cugb.edu.cn (J.C.)

**Keywords:** graphdiyne, nitrogen-doped, nanotube, oxygen reduction reaction, electrocatalyst

## Abstract

Electrocatalysts with high efficiency and low cost are always urgently needed for oxygen reduction reaction (ORR). As a new carbon allotrope, graphdiyne (GDY) has received much attention due to its unique chemical structure containing sp- and sp^2^-hybridized carbons, and intrinsic electrochemical activity ascribed to its inherent conductivity. Herein, we prepared two graphdiyne materials named GDY nanotube and nitrogen-doped GDY (NGDY) nanotube via cross-coupling reactions on the surface of Cu nanowires. As metal-free catalysts, their electrocatalytic activities for ORR were demonstrated. The results showed that the NGDY nanotube presents more excellent electrochemical performance than that of the GDY nanotube, including more positive potential and faster kinetics and charge transfer process. The improvement can be ascribed to the greater number of structural electrocatalytic active sites from nitrogen atoms as well as the hollow nanotube morphology, which is beneficial to the adsorption of oxygen and acceleration of the catalytic reaction. This work helps develop high-quality graphdiyne-based electrocatalysts with well-defined chemical structures and morphologies for various electrochemical reactions.

## 1. Introduction

Nowadays, different forms of environmental degradation have a significant negative impact on human life. With the increasing energy demand, it is extremely important to develop highly efficient energy storage and conversion technology [1,2]. As new types of clean energy devices, fuel cells (such as metal–air batteries and hydrogen fuel cells) have received a lot of attention and achieved significant progress in recent decades [3,4,5]. Oxygen reduction reaction (ORR) is an important process during the energy storage and conversion of the cells [6,7]. Usually, platinum is effective in the electrocatalysis of ORR [8], but the high cost and low reserve restrict its commercialization in high-energy storage devices such as metal–air batteries [9,10]. Therefore, for ORR, there is an urgent need to develop alternatives to replace the noble metal. Due to their high conductivity and perfect stability, carbon materials (such as graphene and carbon black) [11,12] have attracted widespread attention from both researchers and industry. However, with the uniform surface charge distribution of sp^2^-hybridized carbons, the traditional carbon materials have no intrinsic catalytic activity and can only be applied as carriers for non-noble metal catalysts in the electrocatalytic process [13,14]. 

Graphdiyne [15], a new kind of carbon allotrope, is composed of benzene rings (sp^2^-hybridized carbons) and diyne links (–C≡C−C≡C−, sp-hybridized carbons). Due to their unique electronic structures and novel triangular molecular pores, GDY and GDY-based materials have shown great potential in the fields of energy storage and conversion [16,17], catalysis [18,19], electronics, and intelligent devices [20,21]. Generally, in electrocatalytic reactions, a large active surface and pores can facilitate the diffusion and benefit the transfer of small ions and molecules [22,23], rendering GDY a highly efficient electrocatalyst in oxygen reduction reaction (ORR) [24], oxygen evolution reaction (OER) [25], hydrogen evolution reaction (HER) [26], overall water splitting (OWS) [27], electrochemical CO_2_ reduction (ECCO_2_R) [28], and nitrogen reduction reaction (NRR) [29]. Most importantly, in contrast to the graphene and carbon nanotube, GDY is synthesized by the cross-coupling reaction of the small precursor. As a result, abundant GDY with various chemical and electronic structures can be easily regulated and prepared. Nitrogen doping is the easiest and most common method used to adjust the electron distribution of the GDY catalyst due to its higher electronegativity. Usually, the common bonding configurations of nitrogen doping are pyridinic nitrogen, pyrrolic nitrogen, and graphitic nitrogen by nitrogen sources (such as NH_3_, aniline, pyridine, pyrrole, or melamine) [30,31,32] or nitrogen-containing precursors [33,34]. There are many reports on the nitrogen doping of graphdiyne materials; however, it is still important to explore other reaction methods and different morphologies of the new carbon allotrope.

In this paper, we successfully prepared GDY and NGDY nanotubes by using Cu nanowires as the substrate and the catalyst for the coupling reactions. The source of the N doping was the precursor pentakis(ethynyl)pyridine (PEP) under mild room temperature rather than nitrogen sources. After removing the Cu nanowires, GDY and NGDY hollow nanotubes were obtained and could be used as metal-free electrocatalysts for the oxygen reduction reaction. The results showed that the NGDY nanotube presents more positive potential and faster kinetics and charge transfer process than those of the GDY nanotube, indicating that N doping in GDY can lead to more structural electrocatalytic active sites and more excellent electrochemical performance [35].

## 2. Results and Discussion

After removing the Cu nanowires in the GDY@Cu and NGDY@Cu (Appendix A), GDY and NGDY hollow nanotubes were obtained. In Figure 1a and Figure 2a, SEM (scanning electron microscope) images of GDY and NGDY show the morphology with higher surface areas than that of traditional GDY powder [15] due to the hollow morphologies, which can provide more space and surface for the adsorption of oxygen [36]. It is worth noting that there were some differences in the surface morphology between the two nanotubes, which may be attributed to the different reactivities of the precursors hexaethynylbenzene (HEB) and pentakis(ethynyl)pyridine (PEP). More acetylenic groups in HEB means higher reactivity [34], resulting in the subsequent growth of GDY nanosheets on the surface of the nanotube. This phenomenon is difficult to manifest in the growth process of the NGDY nanotube, despite the increase in the amount of the precursor PEP. This can also be demonstrated by TEM (transmission electron microscope) measurement (Figure 1b and Figure 2b). In the high-resolution TEM (HRTEM) images, the GDY and NGDY nanotubes have well-defined lattice fringes in Figure 1c and Figure 2c, with interlayer distances of approximately 0.35 and 0.39 nm (Appendix A), well consistent with the published GDY material [37,38]. Furthermore, the element distribution of GDY and NGDY nanotubes was also confirmed by energy dispersive spectrum (EDS) elemental mapping. In Figure 1d and Figure 2d, the uniform distribution of carbons in the GDY nanotube, and carbon and nitrogen elements in the NGDY nanotube can be seen, indicating successful intrinsic N doping in the NGDY nanotube. Compared to carbon elements, nitrogen has greater electronegativity, and nitrogen doping can influence electrochemical behavior [39,40]. Thus, in this work, the nitrogen-doped graphdiyne nanotube may have more advantages as a metal-free catalyst of oxygen reduction reaction due to its superior electrochemical activity compared to that of the graphdiyne nanotube.

The structures of GDY and NGDY nanotubes were investigated by Raman spectra. In Figure 3a, the two nanotubes showed an obvious D band (around 1390 cm^−1^, E_2g_ stretching vibration) and G band (around 1587 cm^−1^, the breathing vibration of sp^2^ carbon), respectively. The defects in the nanotubes can be evaluated via the relative strength ratio (I_D_/I_G_). Doping with other elements or modified by substituent groups are common methods to produce defects. After N doping, the material pore size was magnified, and the defect was increased. In detail, the defects in GDY and NGDY were calculated to be 0.54 and 0.72, respectively, indicating that the NGDY nanotube possessed more structural defects than the GDY nanotube. Importantly, the Raman peak at around 2159 cm^−1^ was attributed to the diyne links (–C≡C–C≡C–) [41], indicating the successful coupling reaction of HEB and PEB on the surface of the Cu nanowires as well as their graphdiyne nature. Figure 3b presents the X-ray diffraction (XRD) patterns of the GDY and NGDY nanotubes. They displayed a broad peak at around 22°, corresponding to the characteristic peak for carbon materials and the TEM interlayer results. XPS (X-ray photoelectron spectroscopy) was used to further analyze the surface chemical composition of the GDY and NGDY nanotubes, which are displayed in Figure 3c. As can be seen in the XPS survey spectrum of the GDY nanotube, there were mainly C 1s and O 1s peaks with amounts of 94.33% and 4.8%. Meanwhile, on the surface of the NGDY nanotube, a N 1s peak can also be observed, with contents of 67.7% (C), 3.24% (N), and 18.81% (O), respectively, in line with the results of the EDS mapping. The difference in oxygen content also demonstrated that the NGDY can absorb more oxygen molecules than the GDY. The high-resolution C 1s XPS spectrum of the GDY nanotube in Figure 3d illustrated four fitted peaks at 283.8, 285.2, 286.5, and 288.9 eV, corresponding to the configurations C-C (sp), C-C (sp^2^), C-O, and O-C=O, respectively [42]. This was also true for the NGDY nanotube, with five fitted peaks at 284.4, 284.9, 285.6, 286.6, and 288.8 eV, corresponding to the configurations C-C (sp), C-C (sp^2^), C=N, C-O, and O-C=O, respectively (Figure 3e) [43]. Figure 3f shows the high-resolution N 1s spectra of the NGDY nanotube, with two peaks attributed to pyridinic N (398.9 eV) and protonic N (400.5 eV) [43]. Heteroatom doping can adjust the electron distribution of carbon-based catalysts. It is generally believed that pyridine N in carbon materials can be applied as active sites to improve the electrochemical performance of electrocatalysts due to their different electronegativities (3.0 for N and 2.6 for C) [35,44]. Therefore, the NGDY nanotube would probably be more effective in catalyzing oxygen reduction reactions than the GDY nanotube.

The electrochemical properties of the GDY nanotube and NGDY nanotube were first evaluated by cyclic voltammetry (CV) in 0.1 M KOH with saturated N_2_ or O_2_ at 10 mV s^−1^. As can be seen in Figure 4a,b and Appendix A, in the N_2_-saturated solution, there were no distinct redox peaks for the two GDY-based materials, while in the O_2_-saturated solution, an obvious cathodic peak can be observed. It is worth noting that the cathodic peak for the NGDY nanotube (0.805 V vs. RHE) was more positive than that of the GDY nanotube (0.707 vs. RHE), demonstrating that the NGDY nanotube showed more perfect catalytic activity. Figure 4c,d show the LSV curves of the GDY nanotube and NGDY nanotube in O_2_-saturated 0.1 M aqueous KOH solution at various rotation speeds from 400 to 1600 rpm. According to Figure 5a, the half-wave potentials of the GDY nanotube and NGDY nanotube were 0.710 and 0.832 V, respectively, confirming that the NGDY nanotube exhibited better ORR performance. On the other hand, there was a kink in the range of 0.75–0.80 V of the LSV curves for the NGDY nanotube, which can be considered as the removal of adsorbed oxygen molecules [45] from the N-doped material during the reaction process due to its large content of oxygen according to the XPS measurements.

According to the Koutecky–Levich (K-L) plots at diverse working potentials in Appendix A, the electron transfer number per O_2_ molecule was calculated and presented in Figure 5b. Furthermore, the RRDE linear sweep voltammogram (Appendix A) was also measured to confirm the electron transfer number (Appendix A). Both measurements indicated that the transfer numbers for the two materials were higher than 3, which means that the ORR was conducted mainly as a four-electron-dominated pathway on the GDY and NGDY electrocatalysts. Accordingly, the production of H_2_O_2_ was lower than 18%. The relatively improved two-electron-dominated pathway of the two GDY nanotubes is different from those reported in the literature [30,46], indicating that the morphology of the materials has an important effect on electrochemical performance. The Tafel slope values were calculated to further demonstrate the reaction dynamics of the two GDY electrocatalysts. As shown in Figure 5c, the Tafel slope values of the GDY nanotube and NGDY nanotube were calculated to be 84.6 and 47.1 mV dec^−1^, respectively, indicating that NGDY showed faster kinetics for electrocatalytic ORR [47]. Furthermore, electrochemical impedance spectroscopy (EIS) was used to determine their ORR behavior after the measurements. The high-frequency semicircles reflect the ion adsorption kinetics and charge transfer resistance (Rct) [48]. As can be seen in the Nyquist plots in Figure 5d and Appendix A, the NGDY nanotube had a smaller resistance than the GDY nanotube. Smaller resistance means faster ion adsorption kinetics and charge transfer. Thus, the NGDY nanotube electrode showed faster ion adsorption kinetics and charge transfer than the GDY one, which benefited the improvement in electrochemical performance.

## 3. Materials and Methods

### 3.1. Preparation of the GDY and NGDY Nanotubes

Cu nanowires were purchased from Nanjing Ji Cang Nano Technology Company (Nanjing, China). The precursors hexakis(ethynyl)benzene (HEB) and pentakis(ethynyl)pyridine (PEP) were prepared according to reported procedures [34]. Before the preparation, Cu nanowires (50 mg) were fully dispersed in ethanol (20 mL) using an ultrasonic disperser. HEB or PEB (10 mg in 40 mL diethyl ether) and pyridine (1 mL) were added to the Cu nanowire solution. The mixture was then refluxed at room temperature for 24 h. After centrifugation, GDY@Cu and NGDY@Cu were obtained. To obtain the GDY and NGDY nanotubes, ammonium persulfate was used to dissolve the Cu nanowires. Then, thermal treatment (150 °C) was used to remove the solvents on the nanotubes in an N_2_ atmosphere before characterization and electrochemical measurement. The synthetic procedure is presented in Figure 1.

### 3.2. Characterization

A Hitachi Model S-4800 field emission scanning electron microscope (Hitachi, Tokyo, Japan) was used to generate the SEM images. A JEM-2100F electron microscope (JEOL, Tokyo, Japan) with an accelerating voltage of 200 kV was used to generate the TEM and HRTEM images. A Renishaw-2000 Raman spectrometer (Renishaw, England, UK) with an Ar laser at 473 nm was used to measure the Raman spectra. An Empyrean diffractometer (Empyrean, London, UK) using Cu_Ka_ radiation with an output power of 1.6 kW at a voltage of 40 kV was used to measure the XRD spectra. A Thermo Scientific ESCALab 250Xi apparatus (EscaLab 250Xi, Waltham, MA, USA) using 200 W monochromated Al_Ka_ radiation was used to measure the XPS spectra.

### 3.3. Electrochemical Measurement

For the oxygen reduction reaction measurement, 70% of the as-prepared GDY and NGDY nanotubes and 25% Super P were uniformly blended with 5% Nafion in ethanol solution. All of the electrochemical performance tests were performed on an electrochemical workstation (CHI760E, Chenhua, Shanghai, China, and RRDE-3A device, ALS, Tokyo, Japan). In the measurement, 0.1 M KOH aqueous solution was used. The cyclic voltammetry (CV), linear sweep voltammetry (LSV) tests, and rotating ring–disk electrode tests were performed in N_2_-saturated electrolyte or O_2_-saturated electrolyte, respectively, with a scan rate of 10 mV s^−1^.

## 4. Conclusions

In summary, graphdiyne and nitrogen-doped graphdiyne nanotubes were successfully prepared on the surface of free-standing Cu nanowires. Benefiting from their hollow morphology and the inherent structure, the as-prepared graphdiyne and nitrogen-doped graphdiyne nanotube were used as metal-free electrocatalysts for the oxygen reduction reaction. After the coupling reaction of the precursor pentakis(ethynyl)pyridine, the well-defined nitrogen configuration and more defects were introduced in the nitrogen-doped graphdiyne nanotube, which is beneficial to the adsorption of oxygen and the acceleration of the catalytic reaction. Thus, the nitrogen-doped graphdiyne nanotube demonstrated superior electrochemical activity for oxygen reduction reaction to that of the graphdiyne nanotube. This work should help develop high-quality graphdiyne-based electrocatalysts with well-defined chemical structures and morphologies for various electrochemical reactions.

## Data Availability

Data are contained within the article and Appendix A.

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
