# Peer review of "Graphdiyne and Nitrogen-Doped Graphdiyne Nanotubes as Highly Efficient Electrocatalysts for Oxygen Reduction Reaction"

_ijms, 2023, doi:10.3390/ijms242316813_

Round 1

Reviewer 1 Report

Comments and Suggestions for Authors

1.     Many grammatical corrections needed.

2.     In line 120, ,” indicating that NGDY nanotube possessed more structural defects than that of GDY nanotube.”

Does not explain the role of structural defects on catalytic activity?

3.     In line 132, “ The difference of oxygen content also demon-strated that the NGDY can absorb more oxygen and oxygen intermediates than GDY” shows the oxidation due to oxygen absorption. How did it work as efficient reductive catalysis?

Statement is controversial.

4.     “It is generally believed that pyridine N in carbon materials can be applied as active sites to improve the electrochemical performance as electrocatalysts [35,44]”

is it for oxidation or reduction?

5.     “ there was a kink in the range of 0.75-0.80 V of the LSV curves for NGDY nanotube, which can be considered as the removal of adsorbed oxygen 160 molecules [45] from the N-doped material during the reaction process due to its large content of oxygen according to the XPS measurements.” Line 159-162.

 Did you compare the XPS data before and after the catalysis?

6.     Did you measure the EIS of the solution or the solid? if it was measured in solution, before or after the experiment? Did you measure solution resistance before the material resistance?

7.     “ As can be seen in the Nyquist plots in Figure 5d, NGDY nanotube electrode showed a faster ion adsorption kinetics and charge transfer than that of GDY, which was benefit to the improvement of the electrochemical performance.” Line 184 and 185.

 The statement will be more illustrative if it adds a sentence “The smaller the resistance the faster ion adsorption kinetics and charge transfer”.

8.     The author may want to write Title number 3 (discussion) as “conclusion”

Comments on the Quality of English Language

needs to improve the language.

Author Response

Dear Editor,

We thank you and reviewers very much for offering us the valuable opportunity and suggestions for improving our manuscript ijms-2714406 entitled “Graphdiyne and Nitrogen-doped Graphdiyne Nanotubes as Highly Efficient Electrocatalysts for Oxygen Reduction Reaction”. We have carefully revised our manuscript according to these comments and your kind suggestions. All the revised parts in this paper are highlighted in red. Followings are our point-by-point responses to these comments.

 Reviewer #1:

Comment 1: Many grammatical corrections needed.

Response: Thank you. According to your suggestion, we have carefully checked our manuscript and the revised parts are highlighted in red.

Comment 2: In line 120,” indicating that NGDY nanotube possessed more structural defects than that of GDY nanotube.” Does not explain the role of structural defects on catalytic activity?

Response: Thank you for your question. According to the published papers (Angew. Chem. Int. Ed. 2023, 62, e202303409; ACS Energy Lett. 2018, 3, 1183), doping with other elements or modified by substituent groups are common methods to produce defects and adjust the electron distribution of carbon-based catalysts. The electronegativity of N is 3.0, higher than that of C (2.6), which can endow adjacent carbon atoms of N-doped sites with electron-withdrawing effect and improve the catalytic activity. This is also true in the graphdiyne material. After N-doping, there are more defects in the NGDY material compared to GDY, which is can be demonstrated by the ID/IG in the Raman spectra. The defects and the non-uniform electron distribution contribute to the improvement of the catalytic activity for NGDY.  

 Comment 3: In line 132, “The difference of oxygen content also demon-strated that the NGDY can absorb more oxygen and oxygen intermediates than GDY” shows the oxidation due to oxygen absorption. How did it work as efficient reductive catalysis?

 Statement is controversial.

Response: Thank you for your comments. In the XPS results, the oxygen contents in the two materials is obviously different. The large amount of oxygen in NGDY can be considered to be containing more oxygen molecules after N-doping with larger molecular pore size. In detail, this phenomenon mainly comes from the adsorption of O2 during the preparation of the GDY with triangular molecular pores (0.6 nm), which is demonstrated by many published literatures (Nat. Commun. 2019, 10, 1165; Adv. Mater. 2019, 31, 1806272; Chem. Commun., 2015, 51, 1834; etc.). In this paper, N-doping from the precursor pentakis(ethynyl)pyridine enlarged the pore size of the material (1.3 nm) (Nano Energy, 2018, 44, 144), indicating more oxygen molecules can be adsorbed. Accordingly, we have improved the description in the revised manuscript.

 Comment 4: “It is generally believed that pyridine N in carbon materials can be applied as active sites to improve the electrochemical performance as electrocatalysts [35,44]”.

Is it for oxidation or reduction?

Response: Thank you for your comments. According to the published papers (Science 2016, 351, 6271, 361-365; Adv. Funct. Mater. 2022, 32, 2204137; J. Phys. Chem. C 2015, 119, 34, 19876-19882), the higher electronegativity of N than C can endow adjacent carbon atoms of N-doped sites with electron-withdrawing effect then increase the catalytic activity. Among the main formation of N doping, pyridinic N is considered to be the most favorable bonding mode for ORR. It is generally known that oxygen molecules can be adsorbed on Lewis base sites (J. Phys. Chem. C 2015, 119, 34, 19876-19882). Carbon atoms next to pyridinic N are considered to be the active sites with Lewis basicity at which O2 molecules are adsorbed as the initial step of the ORR.

Comment 5: “there was a kink in the range of 0.75-0.80 V of the LSV curves for NGDY nanotube, which can be considered as the removal of adsorbed oxygen 160 molecules [45] from the N-doped material during the reaction process due to its large content of oxygen according to the XPS measurements.” Line 159-162. Did you compare the XPS data before and after the catalysis?

Response: Thank you for your question. For the oxygen reduction reaction measurement, 70 % of the as-prepared GDY and NGDY nanotubes, 25% Super P were uniformly blended with 5 % Nafion in the ethanol solution. As a result, it is hard to compare the XPS results before and after the ORR tests.

Comment 6: Did you measure the EIS of the solution or the solid? if it was measured in solution, before or after the experiment? Did you measure solution resistance before the material resistance?

Response: Thank you for your question. Actually, the EIS was measured after the electrochemical experiment by the electrochemical workstation. According to this suggestion, we have supplemented the resistance data of the solution (0.1 M KOH aqueous solution) before the test in the revised supporting information.

Comment 7: “As can be seen in the Nyquist plots in Figure 5d, NGDY nanotube electrode showed a faster ion adsorption kinetics and charge transfer than that of GDY, which was benefit to the improvement of the electrochemical performance.” Line 184 and 185.

The statement will be more illustrative if it adds a sentence “The smaller the resistance the faster ion adsorption kinetics and charge transfer”.

Response: We have improved our description according to your suggestion in the revised manuscript.

Comment 8: The author may want to write Title number 3 (discussion) as “conclusion”

Response: Thank you. We have corrected the mistake in the revised manuscript. Thank you again for your kind help in the improvement of the paper.

 Reviewer #2: This paper explores two types of electrocatlysts for oxygen reduction reaction. The authors present clear results backed by supportive evidence. I recommend a minor revision before publication. Detailed comments are as follows:

Comment 1: The authors used N-doped graphdiyne for improved ORR electrocatalyst. As the authors mentioned, several images of N-doped graphdiyne have larger surface area compared to graphdiyne. It might be enough to prove the claim, but for the electrochemical activity the author can measure the electrochemically active surface

area. Could you please provide the improvement of the surface area by electrochemical method?

Response: Thank you for your suggestion. According to your advice, we have measured the electrochemically active surface area by CV curves. As can be seen in Figure S11 in the revised supplementary materials, NGDY provides a higher electrochemical active area than that of GDY, which can expose enough active sites and reduce steric hindrance and mass transfer resistance. As a result, NGDY nanotube shows more excellent electrochemical kinetic behavior.

Comment 2: In lines 104, “Fig-ure” should be corrected to “Figure”.

Response: We have made corrections in the revised manuscript.

Comment 3: Many sentences of third paragraph in introduction part is same with several sentences of abstract. Please revise one of them.

Response: Thank you. We have improved our description in the revised manuscript and the revised parts were highlighted in red.

Thank you very much.

Sincerely yours!

Hong Shang, Associate Professor

Xuanhe Liu, Associate Professor

School of Science

China University of Geosciences (Beijing), Beijing 100084, P. R. China

E-mails: shanghong@cugb.edu.cn; liuxh@cugb.edu.cn

Tel.: 86-10-82322758

Reviewer 2 Report

Comments and Suggestions for Authors

I attached the file. 

Comments on the Quality of English Language

Author Response

Reviewer #2: This paper explores two types of electrocatlysts for oxygen reduction reaction. The authors present clear results backed by supportive evidence. I recommend a minor revision before publication. Detailed comments are as follows:

Comment 1: The authors used N-doped graphdiyne for improved ORR electrocatalyst. As the authors mentioned, several images of N-doped graphdiyne have larger surface area compared to graphdiyne. It might be enough to prove the claim, but for the electrochemical activity the author can measure the electrochemically active surface

area. Could you please provide the improvement of the surface area by electrochemical method?

Response: Thank you for your suggestion. According to your advice, we have measured the electrochemically active surface area by CV curves. As can be seen in Figure S11 in the revised supplementary materials, NGDY provides a higher electrochemical active area than that of GDY, which can expose enough active sites and reduce steric hindrance and mass transfer resistance. As a result, NGDY nanotube shows more excellent electrochemical kinetic behavior.

Comment 2: In lines 104, “Fig-ure” should be corrected to “Figure”.

Response: We have made corrections in the revised manuscript.

Comment 3: Many sentences of third paragraph in introduction part is same with several sentences of abstract. Please revise one of them.

Response: Thank you. We have improved our description in the revised manuscript and the revised parts were highlighted in red.

Thank you very much.

Sincerely yours!

Hong Shang, Associate Professor

Xuanhe Liu, Associate Professor

School of Science

China University of Geosciences (Beijing), Beijing 100084, P. R. China

E-mails: shanghong@cugb.edu.cn; liuxh@cugb.edu.cn

Tel.: 86-10-82322758

Round 2

Reviewer 1 Report

Comments and Suggestions for Authors

The response for the comment 2 and comment 5 should be mentioned in the manuscript to make the readers clear. The authors did not mentioned it that they have been added to the manuscrpt.

Author Response

Dear Reviewer,

Thank you for your comments. Please see the attachment (the revised manuscript was added in the "author-coverletter-33549939.v2.docx"). According to your suggestion, we have added corresponding description in the characterization as:

“Doping with other elements or modified by substituent groups are common methods to produce defects. After N-doping, the material pore size was magnified and the defect was increased.”

“Heteroatom doping can adjust the electron distribution of carbon-based catalysts. It is generally believed that pyridine N in carbon materials can be applied as active sites to improve the electrochemical performance of electrocatalysts due to their different electronegativities (3.0 for N and 2.6 for C).”

“4.3. Electrochemical measurement” as “For the oxygen reduction reaction measurement, 70 % of the as-prepared GDY and NGDY nanotubes, 25% Super P were uniformly blended with 5 % Nafion in the ethanol solution.”

Thank you very much.

Sincerely yours!

Hong Shang, and Xuanhe Liu
